# Preoperative Magnetic Resonance Imaging as a Diagnostic Aid for Hypermobile Lateral Meniscus

**DOI:** 10.3390/diagnostics11122276

**Published:** 2021-12-05

**Authors:** Seikai Toyooka, Naoya Shimazaki, Hironari Masuda, Noriaki Arai, Wataru Miyamoto, Shuji Ando, Hirotaka Kawano, Takumi Nakagawa

**Affiliations:** 1Department of Orthopaedic Surgery, Teikyo University School of Medicine, Tokyo 173-8606, Japan; blueocean1092@hotmail.com (S.T.); hironari@fj9.so-net.ne.jp (H.M.); no_3_noriaki_0729@yahoo.co.jp (N.A.); miyamoto@med.teikyo-u.ac.jp (W.M.); hkawano-tky@umin.net (H.K.); 2Department of Orthopaedic Surgery, Shimazaki Hospital, Tokyo 317-0076, Japan; naoyas@dream.com; 3Department of Information Engineering, Tokyo University of Science, Tokyo 162-8601, Japan; shuji.ando@rs.tus.ac.jp

**Keywords:** magnetic resonance imaging, hypermobile lateral meniscus, popliteomeniscal fascicle, popliteal hiatus

## Abstract

Background: Hypermobile lateral meniscus is difficult to diagnose with imaging due to its absence of tears or anomalies. We aimed to clarify the accuracy of the preoperative diagnosis using magnetic resonance imaging (MRI). Methods: The preoperative MRI status of the posterosuperior popliteomeniscal fascicle (sPMF), anteroinferior popliteomeniscal fascicle (iPMF), and popliteal hiatus were examined retrospectively on sagittal images in the hypermobile lateral meniscus group (*n* = 22) and an age- and gender-matched control group (*n* = 44). These statuses were evaluated by a logistic regression analysis to assess their degree of diagnostic accuracy. Results: The area under the curve (AUC) of the sPMF, iPMF, popliteal hiatus, and all three criteria combined was 0.66, 0.74, 0.64, and 0.77, respectively (low, moderate, low, and moderate accuracy, respectively). The odds ratios of the most severe type 3 forms of the sPMF, iPMF, and popliteal hiatus for hypermobile lateral meniscus were significantly high (5.50, 12.20, and 5.00, respectively). Although the diagnostic accuracy was not high enough, the significantly higher odds ratio for type 3 may indicate a hypermobile lateral meniscus. Conclusion: a definitive diagnosis of hypermobile lateral meniscus is difficult with MRI findings alone; however, MRI evaluations of the iPMF, sPMF, and the widening of popliteal hiatus can be used as an adjunct to diagnosis.

## 1. Introduction

Hypermobile lateral meniscus is characterized by a history of painful catching or locking symptoms and is diagnosed arthroscopically by reproducing the subluxation of the lateral meniscus without tears or anomalies [1]. Once a definitive diagnosis is made, a meniscus suture can be used to treat the condition without symptoms in most cases [2,3,4,5,6,7,8,9,10]. The hypermobile lateral meniscus is difficult to diagnose using imaging because there is no tear or morphological abnormality of the meniscus itself and it must be diagnosed based on clinical findings alone [10,11]. Although good results can be achieved if the disease is correctly diagnosed, many orthopedic surgeons find it difficult to diagnose; therefore, many patients affected by their symptoms may not receive appropriate treatment.

Recent studies have reported that the posterosuperior popliteomeniscal fascicle (sPMF) and the anteroinferior popliteomeniscal fascicle (iPMF) contribute significantly to the stability of the posterior lateral meniscus [12,13]. These are the fibers that connect the posterior part of the lateral meniscus to the joint capsule, and are located on the medial and lateral sides of the popliteal tendon. These structures can be visualized with an arthroscope [10,14]. Since these PMFs are the primary restraints to anterior displacement of the posterolateral corner of the lateral meniscus at 90 degrees of flexion of the knee joint, when these fibers tear or stretch out for some reason, it is thought to cause hypermobile lateral meniscus [10,11]. It has been reported that such important fibers can also be depicted in the sagittal view of magnetic resonance imaging (MRI) [1,10,11,15]. Although there is a study which stated that there is a high incidence of a widening popliteal hiatus as a result of injury to the sPMF and iPMF in patients with hypermobile lateral meniscus [11], there are no reports so far on the diagnostic accuracy of evaluating PMFs by MRI. Therefore, the purpose of this study is to clarify the diagnostic accuracy of MRI for hypermobile lateral meniscus. If the diagnostic accuracy of MRI can be confirmed to be high, it could potentially serve as a diagnostic aid for many orthopedic surgeons.

## 2. Materials and Methods

### 2.1. Patients and Design

Patients who underwent arthroscopic surgery by a single surgeon at two institutions between December 2016 and June 2022 were examined. The study population were divided into the following two groups: the hypermobile lateral meniscus group (Group H) consisted of patients who were diagnosed based on clinical findings, presented a history of mechanical locking or catching episodes with pain along the lateral joint line, exhibited a posterior segment of the meniscus that subluxated anterior to the apex of the lateral femoral condyle during arthroscopic probing of the knee at 90 degrees flexion, and showed no tears or abnormal morphology of the meniscus under arthroscopy. The control group (Group C) consisted of those who underwent arthroscopy by the same surgeon during the same time period and had no abnormal mobility, tears, or morphological abnormalities of the meniscus under arthroscopy. Group C was matched for age and gender to Group H. After arthroscopic treatment in both groups, the MRIs of both groups were evaluated retrospectively.

The study protocol was approved by the institutional review board of the authors’ institutions and all patients provided informed consent. The exclusion criteria were patients with apparent instability of knee joints, and patients with lateral osteoarthritis of the knee joints of grade 2 or higher on the Kellgren–Lawrence grading system on radiographic examination, because lateral osteoarthritis may affect the magnetic resonance imaging (MRI) findings of PMFs by inducing inflammation around the PMFs.

### 2.2. MRI Measurement

Coronal, sagittal, and axial images were obtained from patients placed in a supine position with their knee in full extension. The images were acquired using an MRI system (MAGNETOM Amira 1.5 T: Siemens, Munich Germany; Discovery MR 3.0 T: GE healthcare, Milwaukee, WI, USA). A series of proton density-weighted images with a 3-mm slice thickness and 256 × 512 matrix were used for sagittal views. A slice that showed the medial border of the fibular head and a good view of the sPMF and iPMF between the popliteus and posterior lateral meniscus was chosen.

The evaluation criteria of this study included the MRI status of the sPMF, the MRI status of the iPMF, and widening of the popliteal hiatus. The status of the sPMF and iPMF on proton density-weighted sagittal MR images was evaluated as Type 1 to 3, as reported by Suganuma et al. [10]. The classification was defined as follows: Type 1, a clear continuous low-signal band from the lateral meniscus to the popliteus tendon; Type 2, no clear or discontinuous bands; Type 3, no bands are visible or the end of band is torn (Figure 1).

Since some studies have reported that the widening of the popliteal hiatus on the MRI leads to a hypermobile lateral meniscus, the opening of the hiatus was also added as an item of evaluation in the current study [11,16]. On proton density-weighted sagittal MRI, Type 1 was defined as no clear high-signal hiatus; Type 2 was defined as a clear high-signal hiatus with a maximum width of 1 mm or less; and Type 3 was defined as a clear high-signal hiatus with a maximum width of 1 mm or more (Figure 2). These criteria were evaluated in a way that did not reveal any patient information.

### 2.3. Statistical Analysis

Interobserver and intraobserver errors were calculated for the type of sPMF, iPMF, and the widening of the popliteal hiatus on MRI by using intraclass correlation coefficients (ICC). To determine the interobserver error, the classifications of these criteria were evaluated by two orthopaedic surgeons who were blinded to clinical features. To determine the intraobserver error, the criteria were evaluated with a 1-week interval between measurements.

The data was analyzed by a statistics expert who selected the analysis method. A logistic regression analysis was used to examine the diagnostic accuracy of sPMF, iPMF, and the widening of the popliteal hiatus for hypermobile lateral meniscus. In addition, the diagnostic accuracy when using all criteria including sPMF, iPMF, and the widening of the popliteal hiatus was also examined using the same method. The objective variable was defined as the presence or absence of hypermobile lateral meniscus, and the explanatory variables were defined as the image type of sPMF, iPMF, or the widening of the popliteal hiatus. The area under the curve (AUC) was evaluated by a receiver operating characteristic (ROC) curve with a logistic regression analysis. The value of AUC was divided into 3 categories: 1.0–0.9 indicated high accuracy, 0.9–0.7 indicated moderate accuracy, and 0.7–0.5 indicated poor accuracy.

The calculation of the sample size based on the method described by Peduzzi et al. [17], indicating that a minimum of 10 cases were required for the hypermobility meniscus group. In addition, the number of cases in the hypermobile lateral meniscus group was set at 22, so that approximately 5 cases belonging to each category could be secured. The number of patients in the control group was set at 44, twice the number of patients in the hypermobility meniscus group, because it was expected that fewer patients would be classified as Type 3 in each criterion.

The level of statistical significance was set at *p* = 0.05, and all calculations were performed using SPSS version 12 (SPSS Inc., Chicago, IL, USA).

## 3. Results

Interobserver errors for the sPMF, iPMF, and the widening of the popliteal hiatus were 0.75, 0.78, and 0.88, respectively. Intraobserver errors for the sPMF, iPMF, and the widening of the popliteal hiatus were 0.89, 0.92, and 0.95, respectively.

Patient characteristics are shown in Table 1. Group H consisted of 22 patients (10 males and 12 females) with a mean age of 31.5 years, all of whom underwent suturing for the lateral posterior horn meniscus using an all-inside meniscal repair device. Group C consisted of 44 patients, double the number of patients in Group H, with a mean age of 30.1 years. The male to female ratio was the same as in Group H. There were 28 cases that underwent anterior cruciate ligament (ACL) reconstruction, 8 cases that underwent medial patellar femoral ligament (MPFL) reconstruction, 4 cases that underwent a medial meniscus suturing, 2 cases that underwent ganglion resection, and 2 cases that underwent plica resection.

In the logistic regression analysis of the sPMF (Figure 3 and Table 2), the odds ratio for having hypermobile lateral meniscus was 2.89 for Type 2 (*p* = 0.093) and 5.50 for Type 3 (*p* = 0.036) when Type 1 was used as a reference. The AUC was 0.658, which was classified as poor accuracy.

In the logistic regression analysis of the iPMF (Figure 4 and Table 3), the odds ratio for having hypermobile lateral meniscus was 4.12 for Type 2 (*p* = 0.057) and 12.20 for Type 3 (*p* = 0.002) when Type 1 was used as a reference. The AUC was 0.737, which was classified as moderately accurate.

In the logistic regression analysis of the widening of the popliteal hiatus (Figure 5 and Table 4), the odds ratio for having hypermobile lateral meniscus was 4.23 for Type 2 (*p* = 0.024) and 5.00 for Type 3 (*p* = 0.034) when Type 1 was used as a reference. The AUC was 0.659, which was classified as poorly accurate. 

In the logistic regression analysis of all three criteria combined (the sPMF, iPMF, and the widening of the popliteal hiatus; Figure 6 and Table 5), the AUC was 0.765, which was classified as moderately accurate.

## 4. Discussion

This study quantitively examined the diagnostic accuracy of MRI findings for hypermobile lateral meniscus. The AUC for the sPMF, iPMF, and the widening of the popliteal hiatus were 0.658, 0.737, and 0.659, respectively. Altogether, the AUC was 0.765. Comparing the three criteria, the iPMF had the highest diagnostic accuracy. When making a diagnosis, the most attention should be paid to the iPMF. Unfortunately, the AUCs were not high enough to provide an accurate diagnosis by MRI findings alone. In addition, despite the high intraobserver errors, the interobserver errors were not high enough for achieving an accurate diagnosis. It may be difficult to diagnose hypermobile lateral meniscus based on initial MRI findings alone. In fact, the iPMF, sPMF, and the widening of the popliteal hiatus sometimes appeared normal even in hypermobile lateral meniscus patients, and vice versa. However, the odds ratio of having a hypermobile lateral meniscus was significantly high if these criteria were Type 3 (the odds ratio for the sPMF was 5.50, *p* = 0.036; the odds ratio for the iPMF was 12.20, *p* = 0.002; and the odds ratio for the widening of the popliteal hiatus was 5.00, *p* = 0.034). Therefore, if these Type 3 criteria on MRI were to be observed preoperatively, the examiner should strongly suspect a hypermobile lateral meniscus. These Type 3 findings may be directly related to the diagnosis and should not be overlooked.

Few studies have evaluated the PMFs using MRI. Suganuma et al. examined the proportions of the sPMF and iPMF in the healthy group and the hypermobile lateral meniscus group [10]. As a result, abnormalities in the sPMF and iPMF (Type 2 and 3) were present in 20% to 40% of the healthy group, and abnormalities in the sPMF and iPMF were present in almost all cases in the hypermobile lateral meniscus group. They stated that the iPMF was a more specific diagnosis because sPMF abnormalities are found at a higher rate in healthy knees than in the iPMF. The results support the current study in that the iPMF had a higher diagnostic accuracy than the sPMF. 

Li et al. reported that the widening of hiatus can be observed in hypermobile lateral meniscus on sagittal and coronal MRI [11]. They reported that the diameter of the popliteus hiatus divided by the tibial plateau in the sagittal and coronal plane was significantly greater in the hypermobile lateral meniscus group. Although we attempted to evaluate the widening of the popliteus hiatus by the same method in this study, the diameter of the hiatus was smaller in our study than in the previous study; therefore, we could not obtain an accurate measurement. In the previous study, there were only 8 cases in the hypermobile group, and there were many cases with a large hiatus. In our study, there were many cases that were difficult to diagnose by imaging because the hiatus was not enlarged, and there were not as many cases with a large hiatus as in previous studies. For this reason, we evaluated the status of the hiatus using a simple method without calculation.

This study demonstrated that preoperative testing using MRI poses limitations for the diagnosis of hypermobile lateral meniscus. Although it is important not to overlook Type 3 findings, clinical findings are still important for diagnosis. Future challenges include how to improve diagnostic accuracy and how not to overlook those who suffer from symptoms. One of the ways to overcome these challenges is to use ultrasonography. There is a growing interest in using ultrasonography in the diagnosis of hypermobile lateral meniscus due to its availability, multiplanar capability, and lower cost compared to MRI. Thanks to its dynamic capabilities for the visualization of superficial structures, ultrasonography can easily show sudden snaps in the peripheral wall of the meniscus when mobilizing the knee [18,19]. However, the accuracy of the diagnosis using ultrasound images varies greatly depending on the proficiency of the person performing the examination, and the images vary depending on the patient’s condition. There are challenges in ensuring that all people are tested with high accuracy. Another way to mitigate this problem is to use MR arthrography. This test, in which a contrast medium or saline is first injected into the joint with subsequent MRI, is mostly used to detect Bankart injuries in patients with suspected repetitive shoulder dislocations. Inflating the joint with a contrast medium or saline can be used to check for a stretched anterior inferior glenohumeral ligament labrum complex (AIGHL) and for joint capsule detachment. In addition to the shoulder, this test may be used to evaluate ligament damage, cartilage damage, and joint capsules in the elbow, carpal, hip, knee, and ankle joints. Some reports have shown that PMFs were accurately delineated using this method [20,21]. Similar to Bankart injuries, where AIGHLs are stretched and the articular capsule is inflated, clinical findings showing stretched PMFs, and an inflated hiatus may accurately assess hypermobile lateral meniscus. Further research is needed to verify this hypothesis.

This study has some advantages. Firstly, this study used the sagittal view instead of the oblique 45-degree coronal view. It was previously reported that the oblique coronal view was preferred for sPMF delineation [22]. The study by Suganuma et al. was conducted under the oblique view along the axis of the popliteus hiatus that was oriented in an anteromedial-to-posterolateral direction with 45-degree oblique coronal images, which is not a standard imaging plane used in imaging routines at many hospitals [10]. However, PMFs have been recently reported to be accurately depicted in the sagittal view as well as the oblique 45-degree view [20]. The results of the present study can be used in the evaluation of MRI, which is commonly performed at many hospitals. Secondly, this study examines the diagnostic accuracy of MRI for those diagnosed using arthroscopy. To date, many studies have used MRI to evaluate the PMFs [1,15]. Few studies have evaluated arthroscopic and MRI results together. Thirdly, an age- and gender-matched control group was used in this study. We adjusted for age precisely because this disease has been reported to be more common in the younger age group of those who are 20 to 30 years old [5].

Despite the aforementioned advantages, this study has some limitations. Firstly, this was a retrospective study, and MRI was evaluated after the diagnosis was made by arthroscopy. Secondly, the number of patients with hypermobile lateral meniscus was small. Due to the small sample size, this study may not have achieved a high diagnostic accuracy. However, the number of patients with this disease is relatively small, and few studies have collected more than 20 subjects. A larger number of these patients were evaluated than previously reported. Thirdly, this study was performed at two different institutes and was evaluated using different MRI systems. However, the imaging conditions were almost the same, and both MRI systems had an accuracy equal to or greater than 1.5 T. In order to minimize MRI errors, the proportion of patients in Group H and Group C at each institute was also matched. Fourthly, this study enrolled patients with different knee problems as controls and this may have affected the results. Patients with an ACL injury were included as part of the control group in this study. It has been reported that the the PMFs may be damaged in some patients with an ACL injury [23,24], and that there is a risk of simultaneous damage to the PMFs during an ACL injury. In the present study, we carefully confirmed that there was no abnormal mobility of the meniscus in the control group prior to their inclusion in the study.

## 5. Conclusions

Although a definitive diagnosis of hypermobile lateral meniscus is difficult with MRI findings alone, an MRI evaluation of the iPMF, sPMF, and the widening of the popliteal hiatus can be used as an adjunct to diagnosis.

## Figures and Tables

**Figure 1 diagnostics-11-02276-f001:**
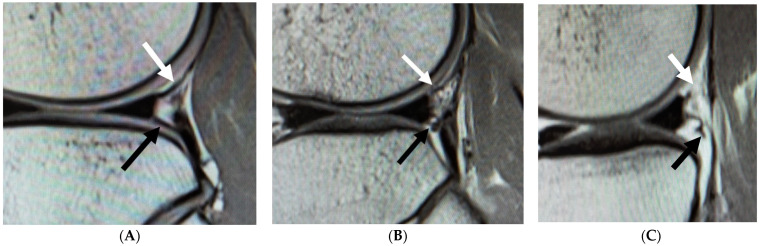
MRI evaluation of sPMF and iPMF. White arrow, sPMF; black arrow, iPMF. (**A**) MRI image of Type 1 sPMF and iPMF in sagittal view. sPMF and iPMF have a clear continuous low-signal band from the lateral meniscus to the popliteus tendon. (**B**) MRI image of Type 2 sPMF and iPMF in sagittal view. There are no clear or discontinuous bands. (**C**) MRI image of Type 3 sPMF and iPMF in sagittal view. There are no visible bands or the end of band is torn.

**Figure 2 diagnostics-11-02276-f002:**
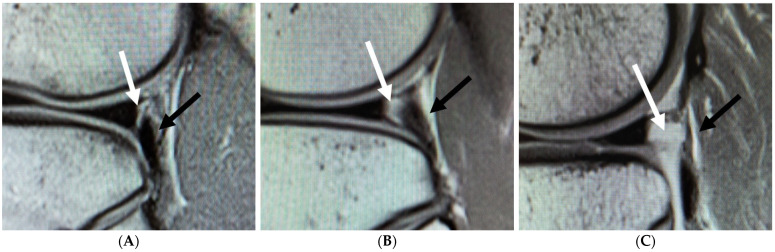
MRI evaluation of the widening of the popliteal hiatus. White arrow, hiatus; black arrow, popliteus. (**A**) MRI image of Type 1 widening of the popliteal hiatus in sagittal view. There is little space between the posterior wall of the lateral meniscus and popliteus. (**B**) MRI image of Type 2 widening of the popliteal hiatus. Type 2 presents a clear high-signal hiatus with a maximum width of 1 mm or less. (**C**) MRI image of Type 2 widening of the popliteal hiatus. Type 3 presents a clear high-signal hiatus with a maximum width of 1 mm or more.

**Figure 3 diagnostics-11-02276-f003:**
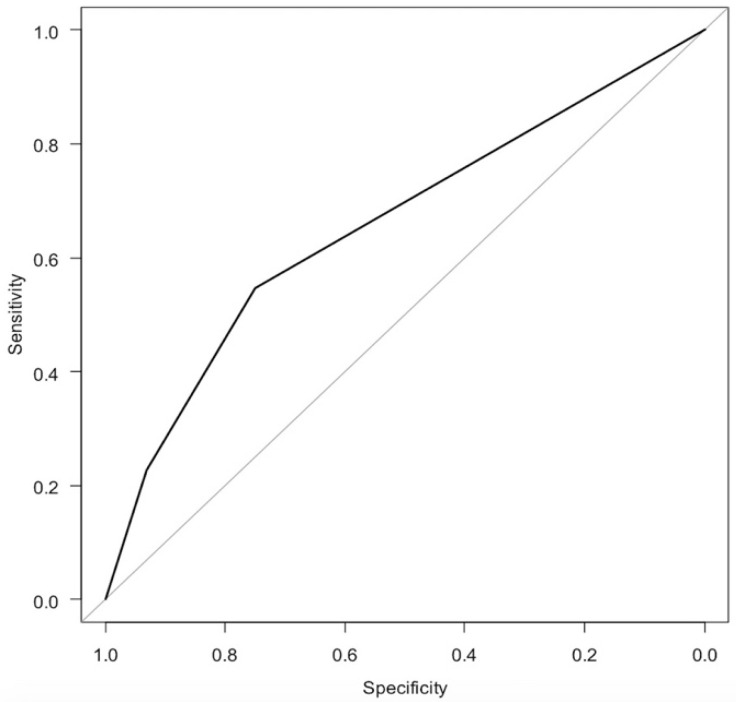
ROC curve of logistic regression analysis of sPMF.

**Figure 4 diagnostics-11-02276-f004:**
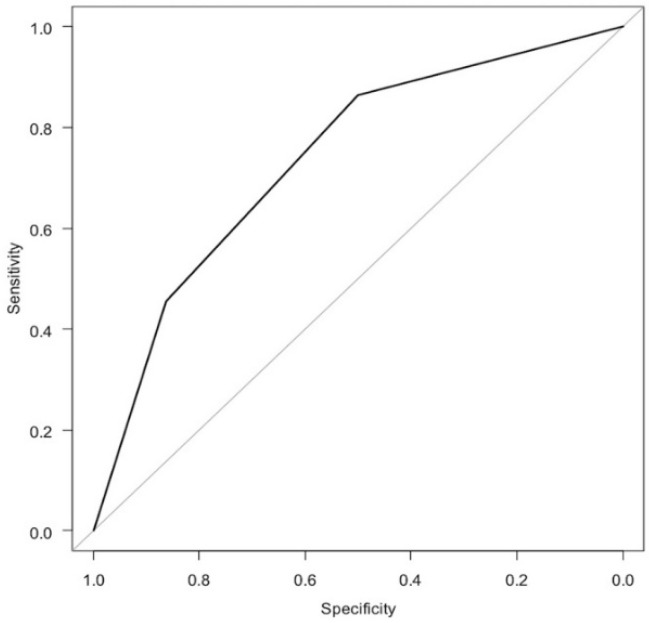
ROC curve of logistic regression analysis of iPMF.

**Figure 5 diagnostics-11-02276-f005:**
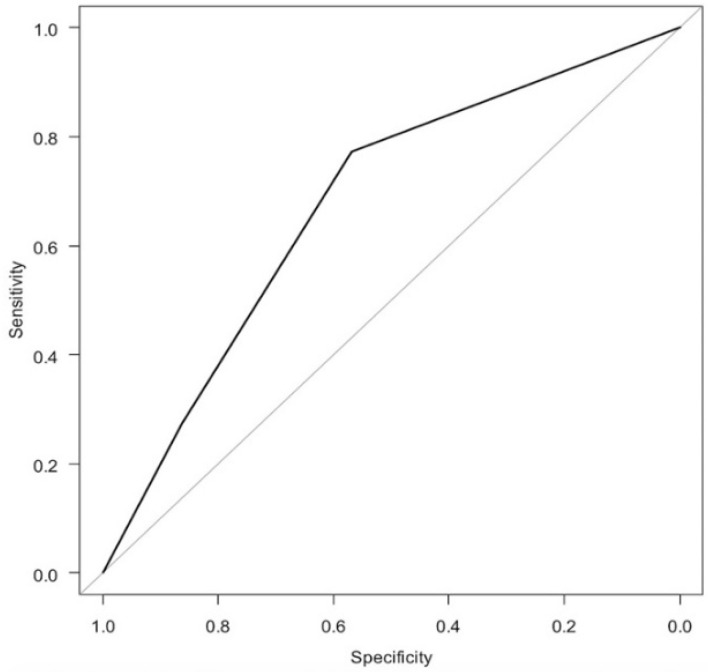
ROC curve of logistic regression analysis of the widening of the popliteal hiatus.

**Figure 6 diagnostics-11-02276-f006:**
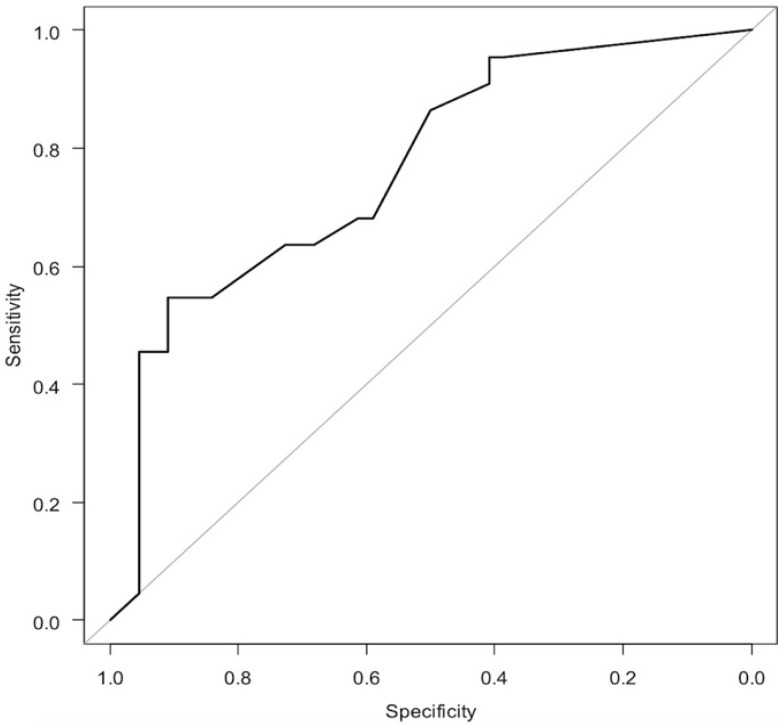
ROC curve of logistic regression analysis of all three criteria combined.

**Table 1 diagnostics-11-02276-t001:** Patient characteristics.

	Hypermobile Meniscus Group (H Group)	Control Group (C Group)
*n*	22	44
Mean age	31.5	30.1
Male:female	10:12	20:24
Operative procedure	Lateral meniscus suture: 22	Anterior cruciate reconstruction: 28
		Medial patellar femoral ligament reconstruction: 8
		Medial meniscal suture: 4
		Ganglion resection: 2
		Plica resection: 2
sPMF	Type 1: 10	Type 1: 33
	Type 2: 7	Type 2: 8
	Type 3: 5	Type 3: 3
iPMF	Type 1: 3	Type 1: 22
	Type 2: 9	Type 2: 16
	Type 3: 10	Type 3: 6
The widening of the popliteal hiatus	Type 1: 5	Type 1: 25
	Type 2: 11	Type 2: 13
	Type 3: 6	Type 3: 6

**Table 2 diagnostics-11-02276-t002:** Logistic regression analysis of sPMF.

	Odds Ratio	95% CI (Lower Limit)	95% CI (Upper Limit)	*p* Value
Intercept	0.303	0.149	0.615	0.001
Type 1 (reference)				
Type 2	2.890	0.838	0.950	0.093
Type 3	5.500	1.110	27.200	0.036
**AUC**		**95% CI** **(lower limit)**	**95% CI** **(upper limit)**	
0.658		0.529	0.786	

CI, confidence interval; AUC, Area Under the Curve.

**Table 3 diagnostics-11-02276-t003:** Logistic regression analysis of iPMF.

	Odds Ratio	95% CI (Lower Limit)	95% CI (Upper Limit)	*p* Value
Intercept	0.136	0.041	0.456	0.001
Type 1 (reference)				
Type 2	4.120	0.961	17.700	0.057
Type 3	12.200	2.530	59.000	0.002
**AUC**		**95% CI** **(lower limit)**	**95% CI** **(upper limit)**	
0.737		0.617	0.856	

CI, confidence interval; AUC, Area Under the Curve.

**Table 4 diagnostics-11-02276-t004:** Logistic regression analysis of the widening of the popliteal hiatus.

	Odds Ratio	95% CI (Lower Limit)	95% CI (Upper Limit)	*p* Value
Intercept	0.200	0.077	0.522	0.001
Type 1 (reference)				
Type 2	4.230	1.210	14.800	0.024
Type 3	5.000	1.130	22.100	0.034
**AUC**		**95% CI** **(lower limit)**	**95% CI** **(upper limit)**	
0.659		0.522	0.796	

CI, confidence interval; AUC, Area Under the Curve.

**Table 5 diagnostics-11-02276-t005:** Logistic regression analysis of all three criteria combined.

	Odds Ratio	95% CI (Lower Limit)	95% CI (Upper Limit)	*p* Value
Intercept	0.110	0.030	0.406	0.001
sPMF Type 1 (reference)				
sPMF Type 2	2.900	0.570	14.800	0.199
sPMF Type 3	6.980	1.040	46.900	0.046
iPMF Type 1 (reference)				
iPMF Type 2	1.130	0.242	5.240	0.881
iPMF Type 3	1.530	0.233	10.000	0.659
Widening of popliteal hiatus Type 1 (reference)				
Widening of popliteal hiatus Type 2	2.070	0.461	9.300	0.343
Widening of popliteal hiatus Type 3	1.910	0.344	10.700	0.459
**AUC**		**95% CI** **(lower limit)**	**95% CI** **(upper limit)**	
0.765		0.644	0.887	

CI, confidence interval; AUC, Area Under the Curve.

## Data Availability

The data presented in this study are available on request from the corresponding author. The data are not publicly available due to privacy restrictions.

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
