# Peer review of "Preoperative Magnetic Resonance Imaging as a Diagnostic Aid for Hypermobile Lateral Meniscus"

_diagnostics, 2021, doi:10.3390/diagnostics11122276_

Round 1

Reviewer 1 Report

The authors present an analysis of preoperative MRI variables as potential diagnostic aids in predicting the presence of a hypermobile lateral meniscus at the time of arthroscopy. 

The main findings of the paper included the higher odds ratio of having a hypermobile lateral meniscus  in the setting of type 3 sPMF, iPMF, and widening of the popliteal hiatus.

I commend the authors for this detailed and comprehensive manuscript investigating a topic not often studied or reported on in the orthopedic literature. 

Below are my suggestions/edits:

Title:

-appropriate

Abstract:

-typo in last sentence of results section in abstract

Introduction:

-comprehensive

Methods:

-comprehensive

Results:

-comprehensive

Discussion/Conclusion:

-comprehensive

Limitations:

-addressed

Reviewer 2 Report

The manuscript entitled “Preoperative magnetic resonance imaging as a diagnostic aid for hypermobile lateral meniscus “ is focused on an interesting topic. The authors tried to use preoperative MRI evaluating posterosuperior popliteomeniscal facicle, anteroinferior popliteomeniscal facicle, and popliteal hiatus to diagnose hypermobile lateral meniscus.

My comments are as follows:

It is not clear how Interobserver and intraobserver errors were calculated. Did the authors use Cohen's Kappa?

The calculation of sample size should be better explained. For example, no effect size was reported.

Line 143: Is 8.8 correct?

Body mass index of the patients was not reported.

In the results, it is not clear how many patients in each group were classified as having Type 1, Type 2 or Type 3 iPMF/sPMF. No descriptive data about widening of popliteal hiatus was reported. Could the authors add a table with these data for each group (H vs C) before presenting logistic regression analysis?

It is not clear what does it mean “Title 1” in table 2, 3, 4 and 5.

Lines 266-293: the authors reported that “First, a larger number of hypermobile lateral meniscus patients were evaluated than previously reported. Since the number of patients  with this disease is relatively small, few studies have collected more than 20 subjects.”. Even if the authors enrolled a larger number of patients compared to other studies, 20 remains a small number of patients. Thus, this point should be moved to the limitations of the study. It is likely that the authors did not find an high diagnostic accuracy because of the small number of patients enrolled.

The authors enrolled patients with different knee problems as controls and this could influence the results. Thus, this point should be reported as a limitation.  

Reviewer 3 Report

The novelty of the study is questionable and is it not clear in which manner this paper adds information if compared to the paper of Suganuma et al.

Furthermore, the results regarding the accuracy of the preoperative MRI are not completely satisfactorily. Indeed, the final message of the authors is that "It may be difficult to diagnose hypermobile lateral meniscus based on initial MRI findings alone."

The auhtors must demostrate how the use of MRI improves the standard diagnostic process. The proposed paper, however, seems just to propose a generic inclusion of MRI that is not well supported by data.

Round 2

Reviewer 3 Report

Although the authors tried to reply to my concerns, I consider their rebuttal not adequate and not exhaustive. I do not think the paper was improved significantly after the review.